# Vitamin D and Calcium Supplementation in Nursing Homes—A Quality Improvement Study

**DOI:** 10.3390/nu14245360

**Published:** 2022-12-16

**Authors:** Charlotte Mortensen, Inge Tetens, Michael Kristensen, Anne Marie Beck

**Affiliations:** 1Department of Nursing and Nutrition, Faculty of Health, University College Copenhagen, 2200 Copenhagen, Denmark; 2Department of Nutrition, Exercise and Sports, Faculty of Science, University of Copenhagen, 1958 Frederiksberg, Denmark; 3Dietetic and Nutritional Research Unit, Herlev Gentofte Hospital, 2730 Herlev, Denmark

**Keywords:** vitamin D, calcium, supplements, nursing homes, adherence, Model for Improvement

## Abstract

Even though dietary supplements with vitamin D and calcium are recommended to nursing home residents, we recently reported a low adherence to this recommendation. The objective of this 20-week quality improvement study was to use the Model for Improvement and Plan-Do-Study-Act (PDSA) cycles to improve adherence in Danish nursing homes. We included two nursing homes with 109 residents at baseline. An information sheet including the rationale for the recommendation was developed for the nurses to urge residents to take the supplements and seek approval by the general practitioner afterwards (PDSA cycle 1). Moreover, it was included in admission meetings with new residents to address supplementation (PDSA cycle 2). A nurse reviewed patient records for number of residents prescribed adequate doses of vitamin D (≥20 µg) and calcium (≥800 mg) before, during and after the intervention. At baseline, 32% (*n* = 35) of the residents had adequate doses of vitamin D and calcium. After implementation of the information sheet and adjustment to admission meetings, this increased to 65% (*n* = 71) at endpoint (*p* < 0.001). In conclusion, in this quality improvement study, we improved the number of prescriptions of adequate doses of vitamin D and calcium over 20 weeks using the Model for Improvement and PDSA experiments.

## 1. Introduction

Despite recommendations of supplementation with vitamin D or vitamin D and calcium to older adults and nursing home residents in a variety of countries, vitamin D deficiency (serum 25-hydroxyvitamin D < 30 nmol/L [1,2,3]) is continuously reported to be widespread among this vulnerable group [4,5,6,7,8]. This can have negative consequences for, physical functioning and bone health, as well as increased risk of falling and mortality [8,9,10,11,12,13]. Recently, our online survey among randomly selected nursing homes in Denmark (*n* = 41) showed that only 8% of nursing homes have a high adherence to the recommendation of giving residents 20 µg vitamin D and 800–1000 mg calcium [14]. A poor adherence was also reported for similar vitamin D and calcium supplement recommendations among nursing home residents in the USA [15,16], Canada [17], Australia [18] and England [19]. Our online survey also showed that the health care professionals’ (HCPs) main reason for not providing the recommended supplements to the residents was that the general practitioners (GPs) did not prescribe vitamin D and calcium to the residents [14]. As the HCPs can only administer the supplements after either a prescription or written consent from the GPs [20], the result is a poor adherence. This is despite the recommendation being a preventive recommendation and not only for treatment [21]. Our data are in accordance with a Belgian survey where 45% of the GPs did not systematically prescribe vitamin D, and, of these, one third only prescribe vitamin D when they remember to [22]. This highlights the need to find strategies to target this underutilization of recommended supplements in long-term care.

The lack of incorporation of a recommendation into routine clinical practice is well-known. Studies report that it may take on average 17 years for evidence-based practice within healthcare to be incorporated into routine clinical practice [23,24]. Thus, there is a need for implementation science, which targets the experienced barriers with strategies considered feasible by the involved HCPs. A widely used method of quality improvement in health care is the Model for Improvement including the Plan-Do-Study-Act (PDSA) cycle, in which small-scale experiments are conducted together with the HCPs to accelerate improvements [25,26,27]. This model has previously been used in two studies targeting low adherence to vitamin D supplementation among nursing home residents. In both studies, adherence improved after 6 [16] and 12 months [17], respectively. However, it has previously been stated that resources are often underestimated when it comes to using the PDSA cycle in quality improvement [28] and that health care studies using the Model for Improvement do often not comply with the key principals of the method [25]. In addition, it has been reported that HCPs find it difficult and time-consuming to follow the method parallel to their primary health care tasks [29]. It could be a barrier for using the Model for Improvement, that it is resource-demanding and leaves less time for primary health care tasks. The objective of this study was to investigate whether using the Model for Improvement in a realistic setting could increase adherence to the vitamin D and calcium supplement recommendation in Danish nursing homes.

## 2. Materials and Methods

### 2.1. Study Design

This was a quality improvement study designed to measure effectiveness, i.e., measure the degree of effect through an intervention conducted under real-world settings. The purpose of the intervention was to increase adherence to the supplement recommendation for nursing home residents. Effectiveness was calculated as the number of prescriptions of vitamin D and calcium to the residents before and after the intervention. The intervention was conducted simultaneously at two Danish nursing homes during 20 weeks from October 2021 to March 2022. No control nursing homes were included. 

### 2.2. Recruitment of Nursing Homes

Two nursing homes in the Region of Zealand, Denmark, were included: nursing home 1 (NH1) and nursing home 2 (NH2). NH1 was recruited based on their participation in our online survey on adherence and barriers to supplementation [14]. NH2 was recruited through use of professional contacts. The inclusion criterion for the nursing homes was an estimated adherence of <40% to the recommendation of residents receiving both supplements, as previously suggested as the definition of low adherence [14].

### 2.3. Baseline Data

Baseline data included number of residents per nursing home, whether a GP was affiliated at the nursing home, and number of residents prescribed adequate doses of vitamin D and calcium (≥20 µg of vitamin D and ≥800 mg calcium, respectively). 

### 2.4. Steps in the Model for Improvement

The three initial steps in the Model for Improvement are shown in Figure 1 [26]. The goal was set for both nursing homes in accordance with the SMART criteria (Figure 1, step 1) [26], as at least 80% of the residents should have both vitamin D and calcium prescribed in at least the recommended doses after a 20-week intervention period. To determine if adherence improved during the intervention, patient journals were reviewed approximately every third week to see if and from when additional residents were prescribed the recommended supplements (Figure 1, step 2). As input before selecting the strategy most suitable to reach the goal (Figure 1, step 3), an interview was performed with each of the involved nurses, and a driver diagram with factors affecting the goal was correspondingly made (Appendix A) [26].

### 2.5. The Plan-Do-Study-Act Cycles

The project’s supervisor and the nurses involved in the project planned the first PDSA cycle. The nurses expressed a need for information about the specific recommendation, e.g., the doses recommended and the health reasons to follow it in order to talk with both residents and their GP about supplementation. Therefore, the project’s supervisor created an information sheet on laminated paper easy for the nurses to bring around to the residents and advise them to have the supplements. Thus, the purpose of this information sheet was to contribute to increased knowledge on the reasoning behind the recommendation of vitamin D and calcium supplements, as this was a prerequisite for the HCPs to be able to argue for supplementation and thereby work for improved adherence. As the recommendation is a general preventive recommendation for all, regardless of vitamin D status and bone health, all residents were targeted this intervention. If a resident agreed to have vitamin D and calcium, the nurse contacted the GP to ensure that supplementation would not interact with any disease conditions or medicine, and the GP correspondingly prescribed the supplements, if appropriate. During PDSA cycle 1, all residents were asked about supplementation (Figure 2). However, to address the turnover of residents, we conducted a PDSA cycle 2 to ensure that new residents were asked to initiate supplementation as early as possible. Therefore, the nurses made a note that all future admission meetings should include urging new residents and their relatives to consider the supplements (PDSA cycle 2).

### 2.6. Effect Evaluation

Approximately every third week, the project’s supervisor contacted the nurse by telephone or attended a meeting (due to the COVID-19 pandemic, most meetings were conducted via telephone). Here, the nurse checked resident journals to see if additional residents had been prescribed vitamin D and calcium supplements, and number of residents and week number were noted by the project’s supervisor. Only residents having ≥20 µg vitamin D and ≥800 mg calcium were noted. Although a written agreement from the GP is sufficient, as vitamin D and calcium are not medicine but dietary supplements [20], the HCPs at both nursing homes only wished to provide supplements if a prescription from the GP was made, and thus the number of prescriptions was chosen as effect evaluation.

### 2.7. Process Evaluation

Process evaluation was planned to include number of residents presented to the information sheet in each week (PDSA cycle 1), as well as numbers of meetings held with new residents and relatives where vitamin D and calcium supplementation was a topic (PDSA cycle 2). 

### 2.8. Statistics

Data were analyzed using the statistical software IBM SPSS Statistics 28.0. Descriptive statistics are presented as numbers and frequencies. McNemar’s test was used to determine if proportions of residents prescribed adequate doses of vitamin D and calcium changed during the intervention from week 1 to 20. A *p*-value of 0.05 determined significance. 

### 2.9. Ethics

The study was conducted in accordance with the Declaration of Helsinki and approved by the Scientific Ethics Committees of The Capital Region of Denmark (H-20031971, 2 December 2020). We anonymized and handled data from the two nursing homes in accordance with the General Data Protection Regulation and the Data protection act. As per the data protection law of 2018, the study was not to be reported to the Danish Data Protection Agency [30]. Data in this paper constitutes a part of data collected at the two nursing homes. Additional data, which will be reported elsewhere, include vitamin D status, muscle strength and physical function in a subgroup of 40 residents. The project is registered at ClinicalTrials.gov as NCT04956705. 

## 3. Results

### 3.1. Baseline Data

Characteristics of the two nursing homes at baseline are shown in Table 1. Initial reviews of patient records by the involved nurses revealed that adherence was below 40% and 30% at NH1 and NH2, respectively. All residents at NH1 had the same GP, as this nursing home had their own GP affiliated. Contrary, no GP was affiliated at NH2, so residents at NH2 had different GPs in the municipality. 

### 3.2. Effect of the Intervention

During the intervention period, four residents (two at each NH) passed away and three new residents moved in (one at NH1 and two at NH2). The primary intervention was the use of the information sheet by the nurses. Collectively, for both nursing homes, adherence increased from around 32% at baseline to 65% at endpoint in week 20 (Table 2), which was a 102.8% improvement (*p* < 0.001). 

In Figure 3, the vitamin D and calcium adherence rate (percentage, %) is shown for each of the 20 intervention weeks at the two nursing homes separately.

### 3.3. Process Evaluation

It was not compatible with a busy day for the nurses to quantify the number of residents who were presented with the information sheet (PDSA cycle 1) on a weekly basis during the intervention. Moreover, in some cases, the information sheet served more as a background-knowledge tool for the nurses in their correspondence with residents rather than a visual tool used in the conversation. When it comes to process evaluation of incorporating the admission meetings to bring up supplementation (PDSA cycle 2), NH1 had one and NH2 had two meetings during the intervention period, at which the residents were encouraged to begin supplementation. This was agreed at all three meetings and supplements correspondingly prescribed by the GPs. Thus, 100% of the meetings with new residents resulted in commencement of supplementation. 

## 4. Discussion

This quality improvement study showed that it is possible to improve adherence to a recommendation of vitamin D and calcium supplementation at nursing homes using the Model for Improvement as a quality improvement method. By use of a simple information sheet about the recommendation and a small adjustment to the admission meetings, adherence at the two included nursing homes increased from baseline values of 38% and 28% of residents having the supplements prescribed to 73% and 59%, respectively. As previously stated, using the Model for Improvement has been reported to be resource-demanding and time-consuming and problematic for the HCPs to follow [25,28,29]. However, we have found that a simple and cheap strategy primarily involving one HCP at each site, who mainly conducted the data collection, and one project supervisor, who created the tool for a simple intervention and assisted with the data collection, can lead to changes in health care. 

Although not based on the Model for Improvement, Munir et al. used a similarly simple approach in an American quality improvement study among 83 long-term care residents. In that study, the medical director at the facility created an educational letter to the GPs with the rationale for calcium and vitamin D supplementation, including the recommended doses. This increased calcium supplementation from 45% to 80% and vitamin D supplementation from 35% to 78% [15]. Thus, the GPs were informed about the benefits of the supplements directly by letter whereas the present study focused on educating the nurses and urged them to contact the GPs (by e-mail or at personal meetings) and argue for an increased adherence to the recommendation. It could be hypothesized that these different communication pathways may explain why adherence was higher in the study by Munir et al. The authors noted, however, making these orders on supplementation a routine on admission for new residents should be considered, which agrees with our strategy on adjustments to admission meetings (PDSA cycle 2). Another American study did use the PDSA cycle approach to improve adherence to vitamin D prescription rates at nursing homes by primarily educating HCPs at a nursing home with a similar sample size (101 residents) and during a similar time frame (five months) as in the present study. In that study, Yanamadala et al., found that vitamin D prescription rate increased from 35% to 86% [16]. The study had a considerably larger project staff compared to the simpler approach with one project supervisor and one involved nurse at each nursing home in our study. Moreover, while the present study only noted prescriptions of both vitamin D and calcium in at least the recommended doses, the study by Yanamadala et al., also counted prescriptions of only vitamin D and vitamin D in multivitamins, which may give higher adherence rates. Tablets containing both micronutrients are larger and may not be prescribed if the resident has chewing-swallowing difficulties. Moreover, some GPs may omit calcium tablets to some residents as they may cause constipation, kidney stones and increased risk of myocardial infarction [9]. However, as the aim of the present study was to evaluate adherence to the national recommendation, only prescriptions in accordance with this were included. 

Other international studies have targeted vitamin D adherence-improvement in larger samples and longer periods. For instance, a Canadian study found that 12 months of knowledge translation strategies based on PDSA cycles in 19 long-term care settings increased vitamin D prescriptions by 22% from a baseline of 36% of residents [17]. Contrary, an Australian study did not find improved vitamin D supplement use after six months of knowledge translation at 17 residential aged care facilities. The authors stated, however, that most of the interventions were not implemented after six months, which could explain the lack of effect [18]. Moreover, one explanation of the different outcomes in the studies could be different baseline prevalence, i.e., 36% had vitamin D prescribed initially in the Canadian intervention homes compared to 57% in the Australian intervention homes. Thus, ours and other studies indicate, that with an initially low adherence of <40%, it is possible to improve prescription rate by targeting the experienced barriers. As we have recently showed that 35% of a representative sample of Danish nursing homes have a low adherence [14], there is a large potential for adherence-improvement in Denmark. 

Despite improved adherence, neither of the nursing homes involved in our study had a high adherence (>80% [16]) at endpoint. NH1 had the highest endpoint adherence of 73%, which could indicate that having a nursing home-affiliated GP is preferable, since it is easier to make several requests for prescriptions at the time. An explanation for why higher adherence was not achieved could be that the residents at the two nursing homes were included in the decision of having the supplements prescribed or not, which was not the case in the studies by Munir et al., and Yanamadala et al. [15,16]. Therefore, even though presented with the positive effects of vitamin D and calcium, some residents refused to have the supplements prescribed. A contributing factor to not reaching a high adherence in the present study could be the general health condition of Danish nursing home residents. The average age of becoming a resident in Danish nursing homes is 84 years old [31] and around one third only live less than one year [32], so most nursing home residents are very vulnerable when moving in. Moreover, approximately 50% of Danish nursing home residents have one or more chronic diseases and approximately 40% are diagnosed with dementia [31]. It could be considered whether strategies other than daily tablets are appropriate for the most vulnerable residents, such as administering higher doses of vitamin D less frequently [33].

This study has some limitations. A major limitation was that we did not include any control nursing homes. Therefore, we cannot know if our intervention caused improved adherence, or if the nursing homes would have increased prescription rates during winter in general. For the Danish adult population in general, vitamin D supplements are recommended from October to March [21] due to lack of cutaneous vitamin D synthesis at this latitude [34], which results in many being vitamin D-deficient during spring [35]. Even though the recommendation for nursing home residents is a general preventive recommendation all year round, it is possible that HCPs and GPs have increased awareness of the recommended supplements during winter and that some improvement in adherence would take place irrespective of an intervention. Based on this, our results may be overestimated and should be interpreted with caution. Another limitation is that the primary outcome was the number of prescriptions of adequate doses of vitamin D and calcium and not the actual consumption of the supplements. As most residents consumed their tablets in their own room after dosage by the HCPs, it was not possible for them to supervise every resident as they consumed the supplements. However, according to the HCPs, prescribed tablets were also dosed, and actual consumption checked regularly by inspection of the dosage boxes. Another limitation is the lack of information on numbers of rejections and reasoning behind rejection of supplementation by either the resident or the GP. This information would be valuable when discussing further strategies to improve adherence. In this regard, it should be mentioned that the present study did not include a third PDSA cycle, as HCPs stated that all residents had already been asked to take a stand, which should be accepted as their final decision from an ethical perspective. The relatively short intervention period of 20 weeks was chosen, as this constitutes the months where no cutaneous vitamin D synthesis occurs at this northern latitude, and since other data collected as part of this study (as vitamin D status measured in a subgroup of residents as presented in Section 2.9) were dependent on lack of cutaneous vitamin D synthesis. Moreover, as previously mentioned, others have found significant improvements with the same study duration as in the present study [16]. However, we cannot rule out that duration of the study or number of PDSA cycles could influence endpoint adherence.

Strengths of the present study include the simple and cheap intervention with limited demands on resources, e.g., documentation. Even though the two involved nurses invested time in the project and its purpose, their participation in the study did not seem to interfere much with routine care tasks, as the chosen intervention could be fitted in. Another strength is the involvement of the relevant HCPs, which ensured that the most relevant barriers were targeted during the intervention. Moreover, as the initial interviews with the nurses confirmed some of the same barriers identified in our previous online survey [14], we find that the targeted barriers are transferable to other nursing homes. 

Another important group to consider is the community-dwelling older adults at 70+ years, as the recommendation of year-round vitamin D and calcium supplementation in Denmark is also targeted at this group and considering that a previous Danish study showed that only 32% of 70+ year-olds still living in own home take the recommended supplements [36]. A higher adherence in this group could automatically improve adherence among the older adults in long-term care, as it will be part of their daily routine before admission. In addition, older adults could stay self-reliant for a longer period as prolonged vitamin D insufficiency reduces bone mineral density and type 2 muscle fibers, leading to increased fragility [37]. Moreover, supplementation with calcium and vitamin D can already reduce bone loss rate and fracture risk in 50+ year-olds [38].

## 5. Conclusions

In conclusion, we have demonstrated that an initially low adherence to the vitamin D and calcium supplement recommendation among Danish nursing home residents can improve within a relatively short period when the nurses are provided with information about the recommendation and make small adjustments to their admission meetings. Our results should, however, be interpreted with caution due to a lack of control nursing homes. Further implementation research projects, preferably including control groups, are needed to evaluate and target experienced barriers among the HCPs. This is not only relevant when it comes to vitamin D and calcium supplementation, but also considering other relevant topics within health care. 

## Figures and Tables

**Figure 1 nutrients-14-05360-f001:**
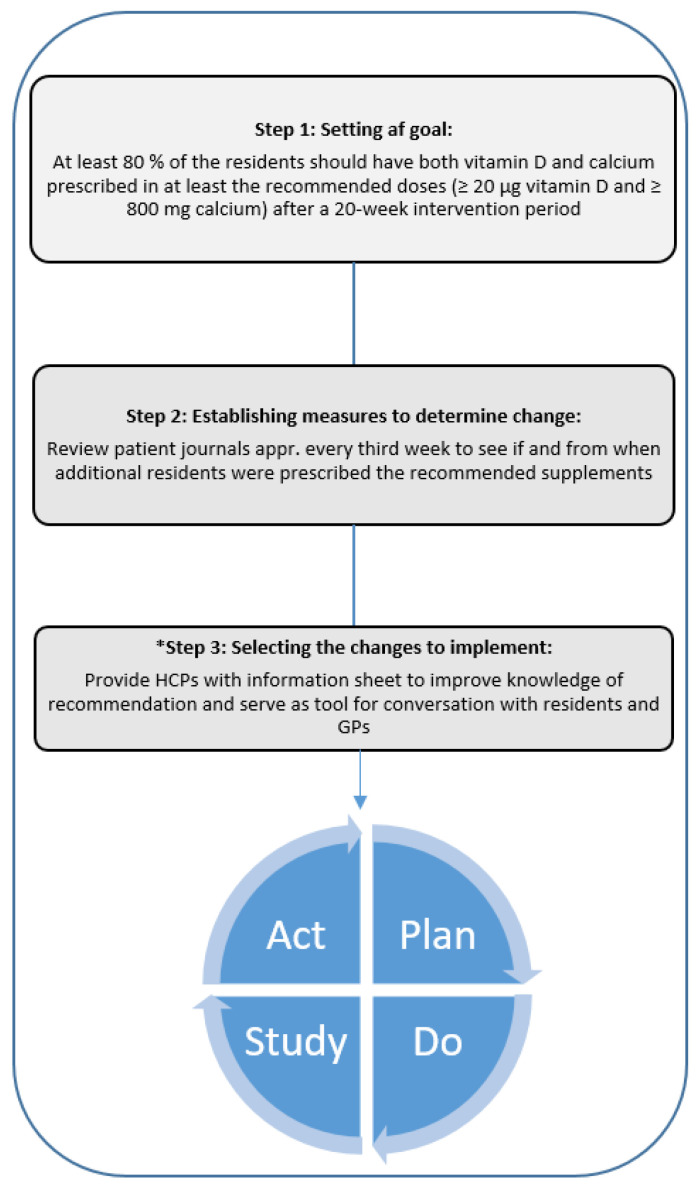
The Model for Improvement. Visual overview of the three initial steps followed by the PDSA cycle. * Input to step three from driver diagram (see Appendix A). Abbreviations: GPs; General practitioner, HCPs; Health care professionals. Figure inspired by [25].

**Figure 2 nutrients-14-05360-f002:**
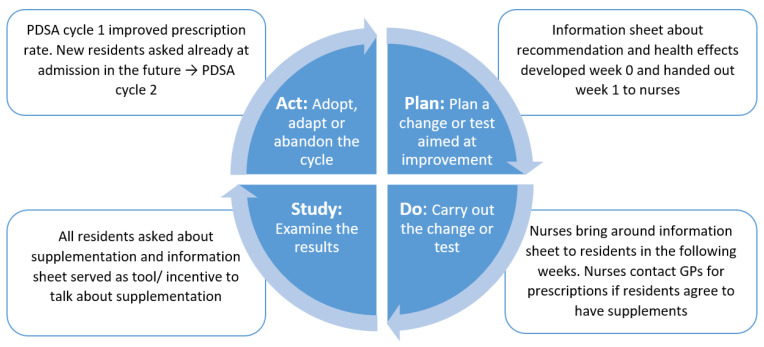
Illustration of the four phases Plan-Do-Study-Act of the PDSA cycle 1 in the project.

**Figure 3 nutrients-14-05360-f003:**
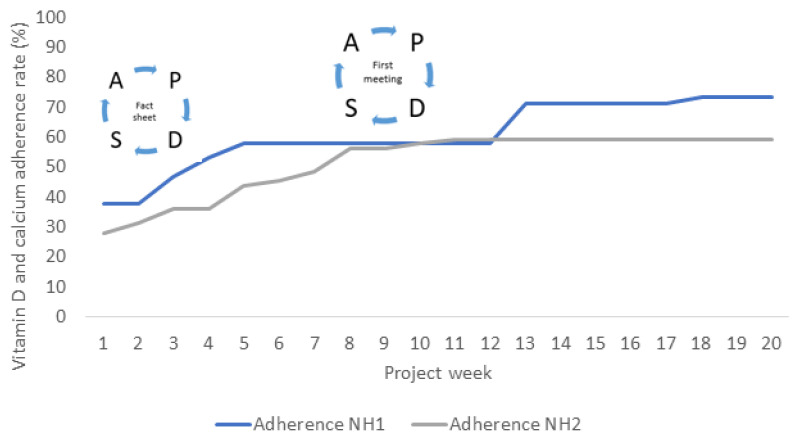
Development of adherence to the recommendation of vitamin D and calcium supplementation (≥20 µg vitamin D and ≥800 mg calcium) at two Danish nursing homes during the 20-week intervention period. “Fact sheet” refers to the information sheet used by the nurses to urge residents to have the supplements, and “first meeting” refers to the adjustment of admission meetings to include supplementation as a topic. Abbreviations: NH; Nursing home, PDSA; Plan-Do-Study-Act.

**Table 1 nutrients-14-05360-t001:** Table of characteristics of the two nursing homes at baseline.

	Nursing Home 1	Nursing Home 2
Number of residents (n)	45	64
Adherence to the recommendation n (%)	17 (37.8)	18 (28.1)
General practitioner affiliated	Yes	No

Adherence was defined as the number of residents prescribed ≥20 µg vitamin D and ≥800 mg calcium after review of resident’s journals.

**Table 2 nutrients-14-05360-t002:** Adherence to the recommendation at the nursing homes at baseline and at endpoint after the 20-week intervention period.

	Baseline Week 1 *n* (%)	Endpoint Week 20 *n* (%)	Week 1–20 ^1^ *p*-Values
NH1 (*n* = 45)	17 (37.8%)	33 (73.3%)	<0.001
NH2 (*n* = 64)	18 (28.1%)	38 (59.4%)	<0.001
NH1 + NH2 (*n* = 109)	35 (32.1%)	71 (65.1%)	<0.001

^1^ Calculated using the McNemar test, statistical significance *p* < 0.05. Adherence was defined as the number of residents prescribed ≥20 µg vitamin D and ≥800 mg calcium after review of resident’s journals. Abbreviations: NH; Nursing home.

## Data Availability

The data presented in this study are available on request from the corresponding author.

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
