# Peer review of "Vitamin D and Calcium Supplementation in Nursing Homes—A Quality Improvement Study"

_nutrients, 2022, doi:10.3390/nu14245360_

Round 1
Reviewer 1 Report
Dear Editors and Authors,
I have taken the time to carefully and thoroughly review the article "Vitamin D and calcium supplementation in nursing homes – a quality improvement study" by Mortensen et al.
This manuscript seems interesting and the topic of discussion is the PDSA Model experiments for Improvement of prescriptions of adequate doses of vitamin D and calcium.
Comments:
·The purpose of this study was to observe the adherence of elderly living in nursing homes to consuming vitamin D and calcium. There are 2 groups, NH1 is willing to participate and NH2 is participating based on the recruitment of health workers.
·The design of this study, does it measure the effectiveness of the intervention program or look at the obedience of the elderly in consuming vitamin D and calcium supplements? it should be clear.
·The involvement of health workers/nurses in this study, is it to increase their competency in seeing the final outcome by preparing tools for them when meeting with the elderly or providing education on the relationship between vitamin D, calcium and the possibility of osteoporosis?
·The program used is the model for improvement and Plan do study act cycle (PDSA) for Danish nursing homes, does it only look at consumption compliance factors without paying attention to the elderly experiencing deficiencies before intervening?
·The baseline was not found and the levels of each group were checked for vit D and calcium, this is important considering that the elderly will be given interventions for vit D and Ca, and of course from the daily diet prepared for the elderly, what is the average content of vit D and calcium subjects? how to explain the success of an intervention if the initial start is not known.
·The fundamental weakness of this study was that it did not have a control that would explain the outcome of the final intervention compared to the NH1 and NH2 groups.
·With reference data per 3 weeks, it can explain whether there is adherence or an increase in consumption of vit D and calcium. A series report (20 weeks/3 weeks) can be made, so there are 6-7 reports based on references that will really help explain the obedience of the elderly.
Please note, the Graphical Abstract or graphical illustration are needed to improve understanding and readability of this manuscript in future publication consideration.
Author Response
Thank you for your valuable review of our manuscript. Below is our point-by-point response to all your comments with our answers set in bold. All amendments are made with track-changes in the manuscript.
Comments
- The purpose of this study was to observe the adherence of elderly living in nursing homes to consuming vitamin D and calcium. There are 2 groups, NH1 is willing to participate and NH2 is participating based on the recruitment of health workers.
RE: It is correct that two nursing homes were recruited. The same intervention was conducted at the two nursing homes. The information on recruitment in section 2.2 of the manuscript explains how contact with the nursing homes involved, were established, i.e., how they were recruited. We have tried to make this clearer by splitting it into two sentences.
- The design of this study, does it measure the effectiveness of the intervention program or look at the obedience of the elderly in consuming vitamin D and calcium supplements? it should be clear.
RE: Thank you for this suggestion to make the text clearer. We have revised section 2.1 to make it clear that the effectiveness of the intervention was evaluated by the number of prescriptions of the recommended supplements.
- The involvement of health workers/nurses in this study, is it to increase their competency in seeing the final outcome by preparing tools for them when meeting with the elderly or providing education on the relationship between vitamin D, calcium and the possibility of osteoporosis?
RE: Thank you for this question. The involvement of the nurses and the tools prepared for them had the main purpose of providing them with enough knowledge about the health effects of the supplements, so they had the competency to argue for and advise residents to have the supplements and afterwards to seek approval from the general practitioner. We have added an explanation of this in section 2.5 to make it clearer.
- The program used is the model for improvement and Plan do study act cycle (PDSA) for Danish nursing homes, does it only look at consumption compliance factors without paying attention to the elderly experiencing deficiencies before intervening?
RE: As the recommendation of vitamin D and calcium supplements to nursing home residents is a general preventive recommendation and not only for deficient older adults, we did not select residents based on for instance vitamin D deficiency but focused on all residents. We have now added a sentence in section 2.5 to make this clearer. Thank you for pointing this out.
- The baseline was not found and the levels of each group were checked for vit D and calcium, this is important considering that the elderly will be given interventions for vit D and Ca, and of course from the daily diet prepared for the elderly, what is the average content of vit D and calcium subjects? how to explain the success of an intervention if the initial start is not known.
RE: The baseline data on intake of vitamin D and calcium from supplements is seen in table 1 and in section 3.1. The mean adherence to the recommendation (“the initial start” with your words) was 32 % of the residents having the supplements at the two nursing homes. Considering your comment, we have added the information on baseline adherence at both nursing homes to the initial part of the discussion. Please see our track-changes in the manuscript. You are correct that we do not know the intake of vitamin D and calcium from the diet. However, according to the recommendation, the supplements are recommended independent of intake from the diet, therefore, irrespective of the amount of vitamin D and calcium from food, all residents are recommended the supplements. In Denmark, very few foods are fortified with vitamin D, and it is almost impossible to reach the recommended intake by diet alone.
- The fundamental weakness of this study was that it did not have a control that would explain the outcome of the final intervention compared to the NH1 and NH2 groups.
RE: We agree that a limitation of the study is the lack of a control nursing home, as we conducted the intervention at both nursing homes. The lack of a control nursing home is in line with two other studies, which had a similar purpose and methodological approach, which we also refer to in the discussion (Munir et al. 2006 and Yanamadala et al. 2012). Even though we do not find it uncommon in quality improvement studies or pragmatic trials not to include a control group, we agree that this limitation should be highlighted and overinterpretation of our results should be avoided. Therefore, we have now mentioned this lack of control nursing home in the beginning of the section instead of in the middle of the section and we have also rephrased parts of the paragraph to acknowledge that some improvement in adherence could take place at a control nursing home due to seasonal changes, although a doubling of prescriptions as seen in this study seems less likely, also considering the very low adherence we showed in our previous online survey (Mortensen, C.; Tetens, I.; Kristensen, M.; Snitkjaer, P.; Beck, A.M. BMC Geriatr. 2022, 22, 27). Also, in section 5 (Conclusion) we have added that future implementation research projects should preferably include control groups. Please see our track-changes in the text.
- With reference data per 3 weeks, it can explain whether there is adherence or an increase in consumption of vit D and calcium. A series report (20 weeks/3 weeks) can be made, so there are 6-7 reports based on references that will really help explain the obedience of the elderly.
 RE: We agree with this comment. Figure 3 shows the increase in adherence to the recommendation during the 20 weeks. Information was gathered every approx. 3 weeks by interviewing the nurses and the exact week from which a resident had the supplements prescribed, were noted.
Please note, the Graphical Abstract or graphical illustration are needed to improve understanding and readability of this manuscript in future publication consideration.
RE: Thank you very much for addressing this important point. We have now made a graphical abstract for the manuscript in order to improve understanding and readability. Please see the uploaded graphical abstract.
Reviewer 2 Report
The manuscript is well-written and addresses an important topic. The major issue is the over-interpretation of findings and conclusions attributed to the intervention since no control group was included. Therefore, the effectiveness of the intervention remains unanswered. Both nursing homes improved similarly (35 percentage points in NH1 and 29 pp in NH2, respectively), but also, is likely that in the winter season there is more awareness about Vit D supplementations and therefore increased their prescription.
No data is presented on the main baseline characteristics of both nursing homes. The authors only mention where the GP was ascribed. Consider adding a table with the main characteristics.
It is not clear why do they choose to evaluate the intervention at week 20? If you are looking for higher adherence, a long-term perspective is desirable. Please, discuss the rationale behind week 20 as a target timeline.
The authors stated that the most relevant barriers were targeted during the intervention but It is not clear why did they not gather data on the consumption of the supplementation?.

Author Response
Thank you for your valuable review of our manuscript. Below is our point-by-point response to all your comments with our answers set in bold. All amendments are made with track-changes in the manuscript.
Comments
The manuscript is well-written and addresses an important topic. The major issue is the over-interpretation of findings and conclusions attributed to the intervention since no control group was included. Therefore, the effectiveness of the intervention remains unanswered. Both nursing homes improved similarly (35 percentage points in NH1 and 29 pp in NH2, respectively), but also, is likely that in the winter season there is more awareness about Vit D supplementations and therefore increased their prescription.
RE: Thank you for your comments on the manuscript and the importance of the topic. We agree that a major issue is the lack of a control nursing home, as we conducted the intervention at both nursing homes. The lack of inclusion of a control nursing home was also the case in two other studies, which had a similar purpose and methodological approach, which we also refer to in the discussion (Munir et al. 2006 and Yanamadala et al. 2012). Even though we do not find it uncommon in quality improvement studies or pragmatic trials not to include a control group, we agree that this limitation should be highlighted and that overinterpretation of our results should be avoided. Because of this, we now mention this lack of control nursing home in the beginning of the section instead of in the middle of the section in the discussion. Moreover, we have also rephrased parts of the section to acknowledge that some improvement in adherence could take place at a control nursing home due to seasonal changes, although a doubling of prescriptions as seen in this study seems less likely, also considering the very low adherence we showed in our previous online survey (Mortensen, C.; Tetens, I.; Kristensen, M.; Snitkjaer, P.; Beck, A.M. BMC Geriatr. 2022, 22, 27). Moreover, we added a sentence in section 5 (Conclusion) that future studies including a control group would be preferred. Please see the document with our track-changes.
No data is presented on the main baseline characteristics of both nursing homes. The authors only mention where the GP was ascribed. Consider adding a table with the main characteristics.
RE: Thank you for this comment. You are correct that our baseline data is limited, and we do not have a Baseline Table. However, we do report on the following baseline data in sections 2.2, 2.3, 3.1 and in Table 1: Geographical region of the nursing home, size in terms of number of residents, whether a GP was affiliated, or if the residents had their own GPs as well as number of residents having the recommended supplements. Based on your comment we have now adjusted section 3.1 Baseline data, to make this baseline data clearer. We do have additional baseline data for a subgroup of 40 residents, which will be published later (as also mentioned in section 2.9). However, due to the general data protection regulation (GDPR) this paper only includes data used directly in the analyses.
It is not clear why do they choose to evaluate the intervention at week 20? If you are looking for higher adherence, a long-term perspective is desirable. Please, discuss the rationale behind week 20 as a target timeline.
RE: Thank you for pointing this out. It is correct that it was not clear in the manuscript why 20 weeks was chosen and that a longer duration could result in a different adherence. We have now added a section on this to the discussion explaining the rationale behind 20 weeks. The study by Yanamadala et al. was also conducted over 5 months (approx. 20 weeks) as the present study and saw an endpoint adherence at 86 %.
The authors stated that the most relevant barriers were targeted during the intervention but It is not clear why did they not gather data on the consumption of the supplementation?
RE: It is correct that the main outcome was the number of prescriptions of the doses and not the actual consumption. However, as also stated in the manuscript, consumption of tablets was checked regularly by the HCPs as the residents used dosage boxes. Because of your request, we have now added a sentence explaining that supervision of consumption was not possible as the residents consumed the tablets in their own rooms.
Round 2
Reviewer 1 Report
The rigorous of this manuscript has been increased through revisions. Thanks.
Author Response
Comments reviewer 1
The rigorous of this manuscript has been increased through revisions. Thanks.
RE: Thank you for your contribution to an improvement of our manuscript.
Reviewer 2 Report
This study is not an effectiveness study. Therefore, the actual effect of the intervention is unknown as they do not have a control group.
No table of characteristics is presented.
Over-interpretation of the findings in discussion attributable to the intervention is still present.
Author Response
Comments reviewer 2
This study is not an effectiveness study. Therefore, the actual effect of the intervention is unknown as they do not have a control group.
RE: We use the term quality improvement study measuring effectiveness based on the following definition: “Effectiveness studies (also known as pragmatic studies) examine interventions under circumstances that more closely approach real-world practice, with more heterogeneous patient populations, less-standardized treatment protocols, and delivery in routine clinical settings.” This is opposed to an efficacy study: “Efficacy studies investigate the benefits and harms of an intervention under highly controlled conditions”. Both definitions are from the paper by Singal et al. 2014: A Primer on Effectiveness and Efficacy Trials: (https://www.ncbi.nlm.nih.gov/pmc/articles/PMC3912314/).
We still agree, though, that the lack of control group is a limitation, and that over-interpretation should be avoided.
Therefore, we have adjusted our discussion on this limitation so it reads as follows:
“This study has some limitations. A major limitation was that we did not include any control nursing homes. Therefore, we cannot know if our intervention caused improved adherence, or if the nursing homes would have increased prescription rates during winter in general. For the Danish adult population in general, vitamin D supplements are recommended from October to March [21] due to lack of cutaneous vitamin D synthesis at this latitude [34] which results in many being vitamin D insufficient during spring [35]. Even though the recommendation for nursing home residents is a general preventive recommendation all year round it is possible that HCPs and GPs have increased awareness of the recommended supplements during winter and that some improvement in adherence would take place irrespective of an intervention. Based on this, our results may be overestimated and should be interpreted with caution”.
Moreover, we have adjusted the final conclusion, so it reads as follows:
“In conclusion, we have demonstrated that an initially low adherence to the vitamin D and calcium supplement recommendation among Danish nursing home residents can improve within a relatively short period when the nurses are provided with information about the recommendation and make small adjustments to their admission meetings. Our results should, however, be interpreted with caution due to lack of control nursing homes. Further implementation research projects, preferably including control groups, are needed to evaluate and target experienced barriers among the HCPs”.
No table of characteristics is presented.
RE: Thank you. We have followed your request and have now added a Table 1 with our baseline characteristics of the two nursing homes.
Over-interpretation of the findings in discussion attributable to the intervention is still present.
RE: Please see our reply to your first comment were we have adjusted the text to avoid over-interpretation.